# BEATS: Optimizing LLM Mathematical Capabilities with BackVerify and Adaptive Disambiguate based Efficient Tree Search

## Abstract

Large Language Models (LLMs) have exhibited exceptional performance across a broad range of tasks and domains. However, they still encounter difficulties in solving mathematical problems due to the rigorous and logical nature of mathematics. Previous studies have employed techniques such as supervised fine-tuning (SFT), prompt engineering, and search-based methods to improve the mathematical problem-solving abilities of LLMs. Despite these efforts, their performance remains suboptimal and demands substantial computational resources. To address this issue, we propose a novel approach, BEATS, to enhance mathematical problem-solving abilities. Our method leverages newly designed prompts that guide the model to iteratively rewrite, advance by one step, and generate answers based on previous steps. Additionally, we employ a pruning tree search to optimize search time while achieving strong performance. Furthermore, we introduce a new back-verification technique that uses LLMs to validate the correctness of the generated answers. Notably, our method improves Qwen2-7b-Instruct's score from 36.94 to 61.52 (outperforming GPT-4's 42.5) on the MATH benchmark. The code is made available at `https://anonymous.4open.science/r/BEATS-A65C/README.md`

## 1 Introduction

LLMs have demonstrated exceptional performance across diverse tasks and domains (Touvron et al., 2023; meta llama, 2024; Bai et al., 2023a), excelling in zero-shot and few-shot scenarios. Recent advancements in scaling laws and fine-tuning have further enhanced their capabilities, enabling their application in complex real-world tasks such as natural language understanding and multimodal processing.

Among the various capabilities of LLMs, mathematical proficiency is crucial, as it reflects not only logical reasoning but also the model's capacity for structured problem-solving. Mastery of mathematical tasks necessitates precision, adherence to complex rules, and the application of algorithms, all of which are essential indicators of an LLM's overall reasoning and cognitive abilities. There are generally two approaches to enhance mathematical capability. The first set of methods trains LLMs to improve their mathematical skills. Models such as Mammoth (Yue et al., 2023; 2024) and Internlm-math (Ying et al., 2024), along with DeepSeek (Shao et al., 2024), utilize vast amounts of data to develop robust mathematical models. The second set of methods employs tree search and self-correction techniques to enhance mathematical abilities. Techniques like ToT (Yao et al., 2024), RAP (Hao et al., 2023), ReST-MCTS* (Zhang et al., 2024), and LiteSearch (Wang et al., 2024) leverage tree structures and search methods such as BFS, DFS and Monte Carlo Tree Search (MCTS). However, both approaches still encounter suboptimal results. They face the following challenges:

**Suboptimal Prompts** Self-improving models (Yao et al., 2024; Wang et al., 2024) typically address problems by either decomposing them into subproblems or rewriting them, followed by solving through methods CoT or Process of Thought (PoT). However, they tend to overlook the issue of ambiguous problem statements. As illustrated by the root node in Figure 1(a), vague expressions can mislead the LLM's understanding.

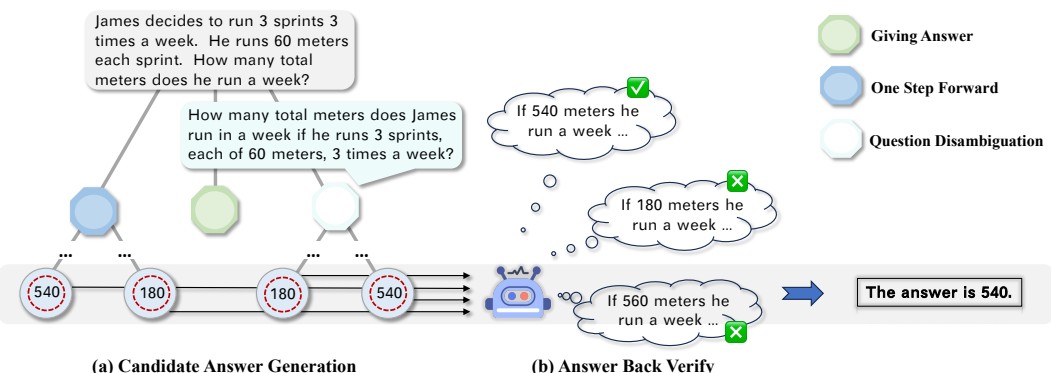

Figure 1: We provide a straightforward example to illustrate our BEATS method. First, we construct a tree search using three distinct actions. Next, we apply back verification to achieve the correct answer.

**High Computational Cost** Previous researches utilizing pre-training or SFT techniques (Yue et al., 2023; 2024; Ying et al., 2024) often suffer from insufficient amounts of data and high computational costs. Search-based approaches enhance mathematical reasoning during the inference stage, thus avoiding the pressure of additional training. However, due to the vast search space, a naive search algorithm can lead to a significant increase in inference time(Yao et al., 2024). Although Wang et al. (2024) employs MCTS to compress the search space, which may result in the absent of correct answers.

**Ineffective Verification Method** When selecting among multiple candidate answers to a problem, previous works like Yao et al. (2024); Wang et al. (2024) typically employ voting-based verification methods. However, they overlook the fact that LLMs can make the same mistakes across multiple routes.

To address these challenges, we propose **BEATS**, a novel method for efficient search aimed at enhancing mathematical performance. Our method guides the model to answer problems instructed by clarified question, thereby avoiding ambiguities in problem statements. We meticulously design prompts that instruct the model to disambiguate, solve one step at a time, and directly generate answers based on preceding steps. Additionally, traditional verification methods in tree search, such as majority voting, may be unreliable, as LLMs can perpetuate the same mistakes across multiple branches. To overcome this, we introduce a back-verification technique that re-submits both the answer and the problem to the model for a judgment of correctness, leveraging the model's capabilities while reducing its reasoning difficulty. Furthermore, we employ a pruning tree search to optimize search time while achieving strong performance. It is worth noting that with our meticulously designed pruning tree, we can control search expenses; simultaneously, compared to MCTS, the pruning tree is able to search through every leaf node, ensuring promising performance, while MCTS is more likely to search based on prior experience.

The core contributions of this paper are summarized as follows:

- **Meticulously Designed Prompt** We developed three newly curated prompts designed to solve mathematical problems step-by-step, provide final answers, and, most importantly, avoiding ambiguities in problem statements.

- **Pruning Tree Search for Controllable Inference Time** We implement a pruning strategy for the tree by imposing constraints on the search steps. Specifically, we restrict the rewriting of the question to once and terminate the tree construction when answer is achieved.

- **New Effective Verification Method** We propose a new back-verification method that re-submits both the answer and the problem to the model for a judgment of correctness, as shown in Figure 1. This approach enhances the performance of searching in LLMs compared to majority voting.

- **Strong Performance** We achieved competitive results across several datasets, including MATH, GSM8K, SVAMP, SimulEq, and NumGLUE. Notably, the BEATS method, based on Qwen2-7B-Instruct, improved its performance on the MATH dataset from 36.94 to 61.52, significantly surpassing GPT-4's score of 42.5.

## 2 RELATED WORK

### 2.1 MATH LARGE LANGUAGE MODELS

LLMs have demonstrated significant capabilities across various tasks, including mathematical problem-solving, which is a critical skill for these models. However, learning to solve mathematical problems poses challenges for LLMs, often requiring large amounts of training data and substantial computational resources. In this paper, we review several state-of-the-art (SOTA) models specifically designed to tackle mathematical problems.

Llemma (Azerbayev et al., 2021) integrates both code and mathematical data to train models, resulting in strong performance. InternLM2 (Ying et al., 2024) utilizes a vast amount of math-related pre-training corpus to achieve high performance. Mammoth (Yue et al., 2023) collected Chain-of-Thought (CoT) data for fine-tuning language models and achieved impressive results. Mammoth2 (Yue et al., 2024) builds on Mammoth by collecting WebInstruct, one of the largest open-source math datasets, and uses it to fine-tune LLMs, resulting in SOTA performance. DeepSeek (Shao et al., 2024) employs preference-based mathematical data to perform an additional stage of reinforcement learning, achieving SOTA results.

In addition to models explicitly trained for mathematics, a few foundation models exhibit exceptional mathematical proficiency. Llama3 (Touvron et al., 2023) has shown remarkable performance in solving mathematical problems. Qwen2 (Bai et al., 2023b), another series of outstanding models, is one of the SOTA open-source models. Furthermore, closed-source models like Claude and GPT also demonstrate strong capabilities in mathematical problem solving.

### 2.2 PROMPT ENGINEERING FOR LARGE LANGUAGE MODELS

The effectiveness of large language models in various applications largely depends on the quality of the prompts used. There are already many designed prompts that can significantly enhance the performance of LLMs (Kojima et al., 2022; Wei et al., 2022; Yao et al., 2024; Besta et al., 2024; Yang et al., 2024; Wang et al., 2023a). However, these methods that rely on manual prompt engineering are far less scalable. In the field of mathematical logical reasoning for LLMs, the Chain of Thought and its derived strategies are widely popular due to their effectiveness. Zero-shot CoT (Kojima et al., 2022) is adding a simple sentence like "Let's think step by step" at the end of questions to assist LLMs in generating reasoning steps. Instead of Zero-shot CoT, Manual-Cot (Wei et al., 2022) provides reasoning steps as few shots. Self-Consistency further improves language models' reasoning performance by generating a diverse set of reasoning paths and choosing the most consistent answer in the final answer set. Tree of Thought (Yao et al., 2024) and GOT (Besta et al., 2024) extend the reasoning pathway from linear to non-linear data structures by leveraging multiple LLM queries to elicit different plausible reasoning paths (Yang et al., 2024). Buffer of Thought (BOT) (Yang et al., 2024) designs a series of thought-template for tasks, and for each problem, it retrieve a relevant thought-template to prompt LLMs. PS prompting (Wang et al., 2023a) improves COT by encouraging LLMs to devise a plan before attempting to solve a problem. In this paper, we employ meticulously designed prompts to enhance the model's mathematical capabilities.

### 2.3 REASONING IN LARGE LANGUAGE MODELS

The recently introduced GPT-o1 model has demonstrated outstanding performance in solving mathematical problems, primarily due to its integration of a novel reasoning module. Our proposed tree search methodology can be categorized as a mathematical reasoning technique. In this paper, we provide a comprehensive review of existing reasoning methods for LLMs. Li et al. Li et al. (2024c) showed that LLMs can achieve arbitrarily high performance with Chain-of-Thought (CoT) prompting. Similarly, Zelikman et al. Zelikman et al. (2024) highlighted the potential for LLMs to "think before reasoning," facilitated by the use of tree structures and verification mechanisms. Tree of

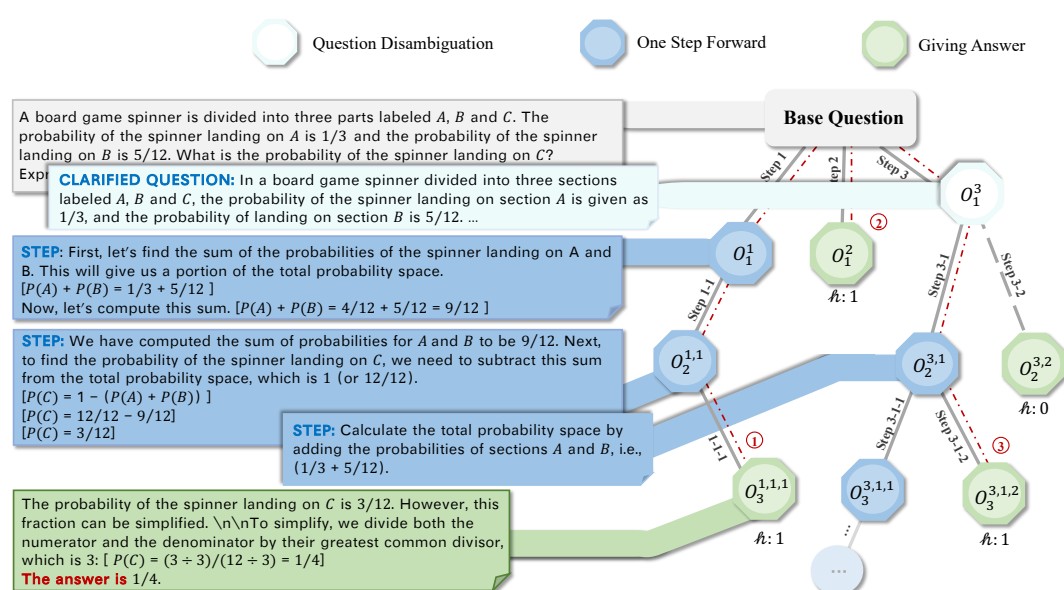

Figure 2: Visualization of the search algorithm in BEATS.

Thought (Yao et al., 2024) leverages tree search and majority voting to improve inference performance. Building on this foundation, Zhang et al. Zhang et al. (2024) applied Monte Carlo Tree Search (MCTS) to achieve efficient and effective tree-based search.

Other works have focused on fine-tuning LLMs to develop self-improvement capabilities. For example, Chen et al. Chen et al. (2024b) employed Step-Level Value Preference Optimization to achieve high model performance. Another related work, AlphaMath Chen et al. (2024a), proposed by Chen et al., utilized value and policy functions along with step-level beam search during inference to enhance mathematical problem-solving abilities. Kumar et al. Kumar et al. (2024) further employed reinforcement learning and oracle feedback to train models for self-correction.

## 3 METHOD

### 3.1 PROMPT DESIGN

We design three actions for the tree search, illustrated in Figure 7. The three options are: One Step Forward, Giving Final Answer, and Disambiguation.

**One Step Forward** The prompt is summarized in Figure 7(a). It encourages the model to progress through the search tree by evaluating the next logical step based on the current context and information. Given that mathematical problems often require multi-step reasoning, splitting a problem into individual steps reduces the complexity of the LLM's response. By addressing each step sequentially, we enhance the likelihood of arriving at the correct answer, as the model can focus on one aspect of the problem at a time, thereby improving accuracy and clarity in reasoning.

**Giving the Final Answer** The prompt is summarized in Figure 7(b), this option directs the model to provide a conclusive answer after considering all relevant information, ensuring clarity and precision in responses. At the appropriate moment, this prompt assists in summarizing the reasoning behind multi-step answers, allowing the model to draw a definitive conclusion. By integrating insights from each step, it helps ensure that the final answer accurately reflects the cumulative logic and reasoning process.

**Disambiguation** The prompt is illustrated in Figure 7(c). This prompt emphasizes reformulating the initial query to enhance clarity and specificity, thereby facilitating a more effective search pro-

cess. This approach is necessary, as many problem descriptions are frequently ambiguous or unclear, leading to incorrect answers. For example, the query, `Josh decides to try flipping a house. He buys a house for $80,000 and then invests $50,000 in repairs. This increased the value of the house by 150%. How much profit did he make?`, can introduce ambiguity. By incorporating a step to rewrite questions, we aim to eliminate such ambiguities, ensuring that the model fully comprehends the problem before attempting to solve it. This helps prevent errors that result from misinterpretations of the initial query.

## 3.2 PRUNING TREE SEARCH

---

**Algorithm 1:** Pruning Tree Building Algorithm

---

**Input:** Maximum depth $D$, question $q$, tree node $u$, action list $A$, one-step action limit $\tau$, LLM generation function $G$, action counter $Count$

**Function** `BuildTree(`$u, d$`)`:

    **if** $d < D$ **then**

        **foreach** $a \in A$ **do**

            **if** $(a = $ `"Disambiguation"`$) \wedge d > 1$ **then**

                **continue**;

            **if** $a = $ `"One Step Forward"` $\wedge Count(u, a) \geq \tau$ **then**

                **continue**;

            $c \leftarrow$ new Node();

            $u.value \leftarrow G(LLM, u.prompt, a)$;

            $c.prompt \leftarrow u.prompt \oplus u.value$;

            $u.addChild(c)$;

            **if** `"the answer is"` $\in c.value$ **then**

                **continue**;

            `BuildTree(`$c, d + 1$`)`;

**Output:** `BuildTree(`$root, 1$`)`

---

In the constructed search tree $\tau$, the root node represents the input question $q$, while the leaf nodes correspond to the deduced answers $S$. The intermediate nodes represent reasoning states that connect the root to the leaves, with edges between these nodes indicating the actions $A$ taken during the reasoning process.

As shown in Figure 2 and Algorithm 1, a node in the tree is denoted by $u_d$, where $d$ indicates the depth of the node. For a given node $u_d$, its ancestor nodes up to the root are denoted by the sequence $u_{d-1}, ..., u_1$. Each node is associated with a prompt that concatenates the responses from previous rounds. These prompts, containing prior rounds of answers, are fed into the action module to generate further responses leading to the correct answer.

$$u_d.\text{prompt} = \bigoplus_{i=1}^{d-1} u_i.\text{value} \tag{1}$$

Additionally, each node stores a value corresponding to the answer derived from both the preceding rounds' responses and the current action. The mathematical formulation is as follows:

$$u_d.\text{value} = G(\text{LLM}, u_d.\text{prompt}, a) \tag{2}$$

We apply the following heuristic pruning rules during this process:

(1) Disambiguation actions are restricted to the immediate successors of the root node to ensure that clarifications or specifications are handled early.

(2) One-step actions are limited to five occurrences within $P_i$, preventing the inference path from becoming excessively long or repetitive.

(3) If a node's content ends with the phrase `The answer is`, the node is marked as a terminal state and added to the set of candidate answers $S$. This rule helps efficiently identify conclusive outcomes, ensuring the search process terminates once a definitive answer is found.

### 3.3 BACK-VERIFICATION

After constructing the tree, we apply a depth-first search (DFS) to identify the leaf nodes. From these, we select only those that contain the phrase `The answer is` as candidate answers for back verification. For a candidate answer $A$, we concatenate it with the question $Q$ for back verification using LLMs:

$$Correct = LLM(Q \oplus A) \tag{3}$$

Back verification involves leveraging both the answer and the question to allow the LLM to confirm the correctness of the answer. It is well-established that verifying an answer is typically easier than solving the original problem. Thus, we employ back verification to enhance the accuracy of validation. After the back-verification, we utilize majority voting based on the back-verification results. The impact of back verification is further examined in Section 4.3.

## 4 EXPERIMENT

Table 1: We compared our method with previous tree search, zero-shot, and SFT approaches on two commonly used benchmarks, i.e. GSM8K and MATH. Our model achieved SOTA performance on both benchmarks.

| | Model | Base Model | Size | MATH | GSM8K |
|---|---|---|---|---|---|
| Zero-Shot | Chain-of-Thought | LLaMA3 | 8B | 27.80 | 50.27 |
| | Chain-of-Thought | Yi-1.5 | 6B | 30.42 | 64.47 |
| | Chain-of-Thought | Qwen2 | 7B | 36.94 | 76.63 |
| | Hard Voting@8 (Wang et al., 2024) | LLaMA3 | 8B | 30.00 | 78.39 |
| | Hard Voting@64 (Wang et al., 2024) | LLaMA3 | 8B | 33.00 | 83.24 |
| SFT | WizardMath (Luo et al., 2023) | LLaMA2 | 7B | 10.70 | 54.90 |
| | MuggleMath (Li et al., 2024b) | LLaMA2 | 7B | - | 68.40 |
| | MetaMath (Yu et al., 2023) | LLaMA2 | 7B | 19.80 | 66.50 |
| | LEMA-LLaMA (An et al., 2023) | LLaMA2 | 7B | 9.40 | 54.10 |
| Search | ToT (Yao et al., 2024) | LLaMA3 | 8B | 13.60 | 69.07 |
| | RAP (Hao et al., 2023) | LLaMA3 | 8B | 18.80 | 80.59 |
| | ReST-MCTS*(1st iteration) | LLaMA3 | 8B | 31.42 | - |
| | ReST-MCTS*(2st iteration) | LLaMA3 | 8B | 34.28 | - |
| | LiteSearch (Wang et al., 2024) | LLaMA3 | 8B | - | 82.30 |
| | Llama-2+M* (BS@16) (Kang et al., 2024) | LLaMA2 | 13B | 32.40 | 66.30 |
| | Llama-2+M* (LevinTS@16) | LLaMA2 | 13B | 33.90 | 68.80 |
| Search | BEATS (w.o. BackVerify) | LLaMA3 | 8B | 35.17 | 83.62 |
| | BEATS | LLaMA3 | 8B | 42.93 | **88.48** |
| | BEATS (w.o. BackVerify) | Yi-1.5 | 6B | 42.01 | 74.68 |
| | BEATS | Yi-1.5 | 6B | 51.27 | 76.12 |
| | BEATS (w.o. BackVerify) | Qwen2 | 7B | 57.28 | 81.50 |
| | BEATS | Qwen2 | 7B | **61.52** | 83.02 |

### 4.1 EXPERIMENT SETTINGS

**Datasets** We conduct experiments on five authoritative mathematical reasoning datasets: (1) **GSM8K**: The GSM8K dataset consists of 1,319 test samples and is widely used for arithmetic problem-solving tasks, designed to evaluate models' performance on grade-school-level math problems. (2) **MATH**: The MATH dataset contains 5,000 test samples drawn from competition-style problems, covering a wide range of topics, including algebra, calculus, combinatorics, and geometry. (3) **SVAMP**: The SVAMP dataset comprises 1,000 math word problems, each involving at

Table 2: We compare our method with previous models on SVAMP, SimulEq, and NumGLUE benchmarks. Our method show significant improvement over these benchmarks.

| | Model | Base Model | Size | SVAMP | SimulEq | NumGLUE |
|---|---|---|---|---|---|---|
| Zero-Shot | Chain-of-Thought | LLaMA3 | 8B | 53.90 | 21.20 | 27.35 |
| | Chain-of-Thought | Yi-1.5 | 6B | 76.40 | 34.63 | 38.39 |
| | Chain-of-Thought | Qwen2 | 7B | 85.20 | 32.68 | 53.36 |
| SFT | Code-Llama (Roziere et al., 2023) | - | 13B | 60.00 | 3.80 | 27.60 |
| | WizardMath (Luo et al., 2023) | LLaMA2 | 13B | 51.90 | 14.90 | 36.10 |
| | Platypus (Lee et al., 2023) | LLaMA2 | 13B | 55.40 | 7.40 | 42.30 |
| | Platypus (Lee et al., 2023) | LLaMA1 | 30B+ | 51.70 | 13.60 | 40.50 |
| | Platypus (Lee et al., 2023) | LLaMA2 | 65B+ | 51.80 | 21.70 | 48.10 |
| | Ocra-Platypus (Lee et al., 2023) | LLaMA2 | 13B | 56.80 | 7.90 | 35.30 |
| | MAmmoTH (Yue et al., 2023) | LLaMA2 | 13B | 72.40 | 43.20 | 61.20 |
| | MAmmoTH-Coder (Yue et al., 2023) | Code-Llama | 13B | 73.70 | 47.10 | 66.40 |
| | Galactica (Taylor et al., 2022) | GAL | 30B | 41.60 | 13.20 | 34.70 |
| | Tulu (Wang et al., 2023b) | LLaMA2 | 30B+ | 59.00 | 10.30 | 43.40 |
| | Guanaco (Dettmers et al., 2023) | LLaMA2 | 65B+ | 66.80 | 20.20 | 40.50 |
| Search | BEATS (w.o. BackVerify) | LLaMA3 | 8B | 80.60 | 72.76 | 66.99 |
| | BEATS | LLaMA3 | 8B | 88.70 | **78.40** | 73.61 |
| | BEATS (w.o. BackVerify) | Yi-1.5 | 6B | 79.30 | 34.72 | 75.43 |
| | BEATS | Yi-1.5 | 6B | 83.70 | 34.82 | **77.93** |
| | BEATS (w.o. BackVerify) | Qwen2 | 7B | 88.80 | 35.21 | 72.84 |
| | BEATS | Qwen2 | 7B | **90.70** | 36.19 | 73.16 |

most two mathematical expressions and one unknown variable. (4) **SimulEq**: The SimulEq dataset includes 514 test samples focused on solving equations, with an emphasis on algebraic manipulation and logical reasoning. (5) **NumGLUE**: The NumGLUE dataset includes 1,042 test problems encompassing 8 distinct tasks that involve various numerical reasoning challenges, such as arithmetic, quantitative reasoning in commonsense and domain-specific contexts, reading comprehension, and natural language inference.

**Models**    To evaluate the effectiveness of our approach, we conducted experiments using three state-of-the-art (SOTA) models: LLaMA3-8B-Instruct, Yi-1.5-6B-Chat, and Qwen2-7B-Instruct. The primary experimental results are presented in Table 1 and Table 2, while a detailed analysis is provided in Section 4.2.

**Baselines**    We consider three types of baseline models: (1) **Zero-Shot Models**, which include Zero-Shot CoT and a hard-voting approach that first generates a set of candidate answers through multiple sampling and then determines the final answer by majority voting. (2) **Supervised Fine-Tuning Models**, encompassing WizardMath (Luo et al., 2023), MuggleMath (Li et al., 2024b), MetaMath (Yu et al., 2023), LEMA-LLaMA (An et al., 2023), Code-Llama (Roziere et al., 2023), Platypus (Lee et al., 2023), MAmmoTH (Yue et al., 2023), Galactica (Taylor et al., 2022), Tulu (Wang et al., 2023b), and Guanaco (Dettmers et al., 2023). (3) **Search Algorithm-Based Models**, including ToT (Yao et al., 2024), RAP (Hao et al., 2023), ReST-MCTS* (Zhang et al., 2024), LiteSearch (Wang et al., 2024), and Llama-2+M* (Kang et al., 2024).

**Details**    In our experimental setup, we configured the tree depth to 7, with the disambiguation step allowed only as a direct successor to the root node. Node expansion was performed using the vLLM framework with the following parameters: temperature set to 0.8, top_p set to 0.9, and max_tokens set to 2048. During the BackVerify stage, Qwen2-7B-Instruct was used as the discriminator. For answer verification, we employed the same framework as MAmmoth. All experiments were conducted on a machine running Ubuntu 22.04, equipped with 8 NVIDIA H100 GPUs, a 120-core CPU, and 960 GB of memory.

## 4.2 MAIN EXPERIMENT

The experimental results presented in Table 1 demonstrate the effectiveness of our proposed method across both the MATH and GSM8K benchmarks. Compared to Zero-Shot category, our model,

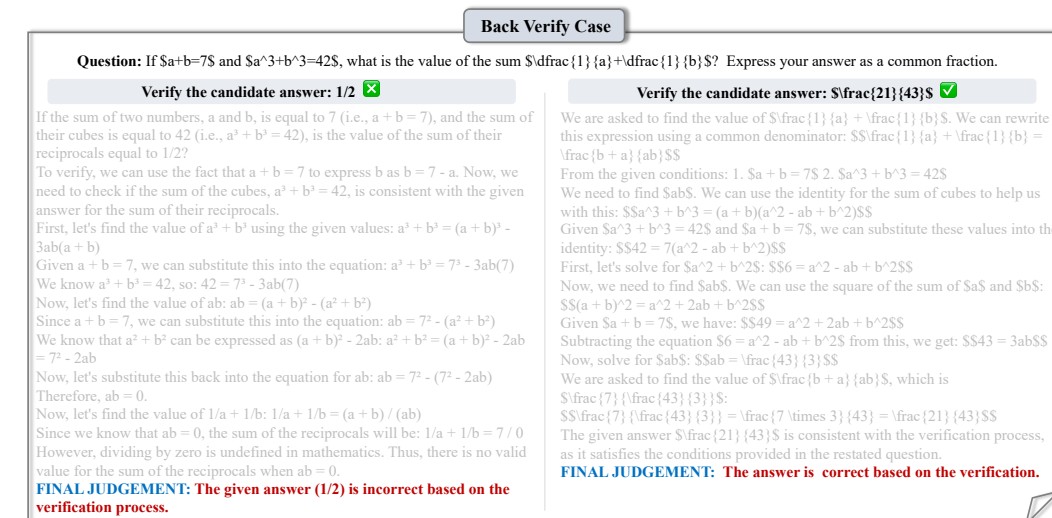

Figure 3: From this figure, we observe that models are more likely to deduce errors when using majority voting but can achieve the correct answer through back verification.

even without the BackVerify step, significantly outperforms these baselines, achieving 35.17% on MATH and 83.62% on GSM8K using LLaMA-8B as the base model. In the Search category, iterative methods like ReST-MCTS* show improvement over time, with the second iteration yielding 34.28% on MATH. Our model, with the BackVerify mechanism enabled, outperforms these methods, reaching 42.93% on MATH and 88.48% on GSM8K with LLaMA-8B. Furthermore, when utilizing the Qwen-7B model, our approach reaches 61.52% on MATH and 83.02% on GSM8K, demonstrating its robustness across different base models. Notably, even without fine-tuning, our approach outperforms the SFT models across both MATH and GSM8K benchmarks. WizardMath and LEMA-LLaMA, both fine-tuned models based on LLaMA-7B, achieve 10.7% and 9.4% accuracy on MATH, respectively, while our method without BackVerify reaches 35.17%, far surpassing the SFT models. Similarly, on GSM8K, WizardMath achieves 54.9% and LEMA-LLaMA reaches 54.1%, whereas our model without BackVerify attains 83.62%, demonstrating a clear performance advantage.

Additional experiments on the SVAMP, SimulEq and NumGLUE datasets consistently prove the effectiveness of our method. On the SVAMP dataset, our model achieves a performance of 88.7 with LLaMA, compared to the best Zero-Shot result of 85.2 using Qwen and the best SFT result of 73.7 from MAmmoTH-Coder. On the SimulEq dataset, our method achieves a significant improvement with a score of 78.4 using LLaMA, outperforming all SFT models, where the highest score is 47.1 by MAmmoTH-Coder. Similarly, on the NumGLUE dataset, our method achieves 73.61, again outperforming both the Zero-Shot and SFT models.

Overall, we have two following observations: (1) Fine-tuning alone may not be sufficient to achieve optimal performance, and that the search-based methods integrated into our approach offer a more robust mechanism for reasoning across tasks. (2) When solving mathematical problems, the MCTS algorithm is not the only viable approach. A straightforward BFS search algorithm, combined with carefully designed long-step and short-step problem-solving prompts along with the BackVerify mechanism, can significantly enhance the model's mathematical capabilities.

### 4.3 ABLATION STUDY

To better understand the strong performance of our model, we conducted an ablation study to demonstrate the effectiveness of the disambiguation and back verification modules by systematically removing them.

**Remove the Disambiguation Module** To assess the impact of the disambiguation process, we conducted a series of comparative experiments using the MATH and GSM8K datasets with both

**Question Disambiguation**

**Original Question:**
James decides to run 3 sprints 3 times a week. He runs 60 meters each sprint. How many total meters does he run a week?

**Clarified Question:**
How many total meters does James run in a week if he runs 3 sprints, each of 60 meters, 3 times a week?

Figure 4: From this figure, we observe that some questions may contain ambiguity, which can be resolved by using the disambiguation operation to generate a clarified version of the question.

the LLaMA3-8b-Instruct and Qwen2-7b-Instruct models. As shown in Table 3, removing the disambiguation component in BEATS resulted in a significant decrease in accuracy across all experiments, highlighting the critical role of the disambiguation process. Additionally, we evaluated the effectiveness of disambiguation through case studies. In Figure 4, the clarified question offers the following advantages: (1) The original phrasing, `"3 sprints 3 times a week"`, is ambiguous, as it could imply that James runs three sprints three times a week or that each session consists of three sets of three sprints. In contrast, the clarified question explicitly states that James runs three sprints per session and completes these sessions three times per week, thereby minimizing potential misinterpretation. (2) The clarified question concisely presents the key details, `"3 sprints of 60 meters each, 3 times a week"`, in a structured format that enhances logical flow and comprehension.

**Remove the Back Verification Module**  In Table 1 and Table 2, we compare model variants with and without back verification across five benchmark datasets: MATH, GSM8K, SVAMP, SimulEq, and NumGLUE. The ablation study demonstrates that back verification consistently improves model performance, highlighting its robustness and effectiveness in enhancing the model's mathematical capabilities. Furthermore, as illustrated by the example in Figure 3, when presented with the candidate answers $\frac{1}{2}$ and $\frac{21}{43}$, the LLM successfully discarded the incorrect solutions through back verification, ultimately selecting the correct answer.

Table 3: We compare the performance with and without the disambiguation module. The results demonstrate the effectiveness of the disambiguation module.

| Dataset | Model | Search | Accuracy |
|---|---|---|---|
| MATH | LLaMA3 | BEATS | 42.93 |
| | | w.o. disambiguation | 35.80 ↓ 7.13 |
| | Qwen2 | BEATS | 61.52 |
| | | w.o. disambiguation | 51.88 ↓ 9.64 |
| GSM8K | LLaMA3 | BEATS | 88.48 |
| | | w.o. disambiguation | 74.83 ↓ 13.65 |
| | Qwen2 | BEATS | 83.02 |
| | | w.o. disambiguation | 76.88 ↓ 6.14 |

Overall, the ablation study demonstrates the critical role of the disambiguation and back verification modules in enhancing model performance. Removing either led to a drop in accuracy, showing their effectiveness in clarifying ambiguous problem statements and filtering incorrect answers. Together, these components significantly improve the model's ability to solve mathematical problems.

## 5 CONCLUSION

In this paper, we introduced BEATS, a new method designed to enhance the mathematical problem-solving capabilities of LLMs. By addressing critical challenges such as suboptimal prompts, ineffective verification methods, and high computational costs, our approach offers a significant improvement in performance. The meticulously crafted prompts facilitate step-by-step reasoning, reducing ambiguities in problem statements and enabling the model to generate accurate answers. Our innovative back-verification technique enhances the reliability of results by ensuring that answers are thoroughly validated. Additionally, the pruning tree search strategy allows for controlled inference time while maintaining state-of-the-art performance. Through extensive experimentation, we demonstrated that BEATS notably outperforms existing methods, marking a solid foundation for advancing mathematical reasoning in LLMs. This work represents an excellent starting point, paving the way for future research to explore more effective verification methods and their applicability across a broader spectrum of complex problem domains.

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

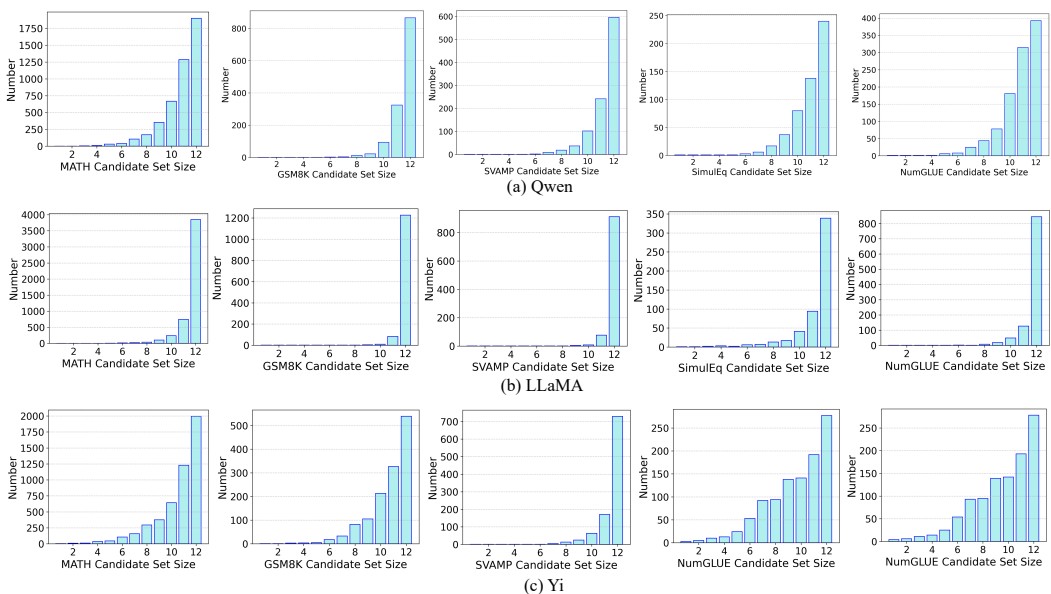

Figure 6: Candidate answer set size.

# A    SEARCH COST AT INFERENCE PHASE

BEATS significantly improves the model's mathematical capabilities through designed pruning search algorithm, which processes multi-turn question inference. Figure 5 presents a comparison of the average number of tokens generated by different models—LLaMA3, Qwen2, and Yi-1.5—across five mathematical benchmarks: MATH, GSM8K, SVAMP, SimulEq, and NumGLUE. As shown in the figure, LLaMA3 consistently produces the highest number of tokens across all benchmarks, with a particularly large margin in the MATH dataset, where it exceeds 5,000 tokens on average. In contrast, Qwen2 and Yi-1.5 generate fewer tokens, with Yi-1.5 often producing the least across most datasets. This suggests that LLaMA3 might engage in more extensive reasoning processes but at the cost of higher computation, while Qwen2 and Yi-1.5 strike a balance between efficiency and performance.

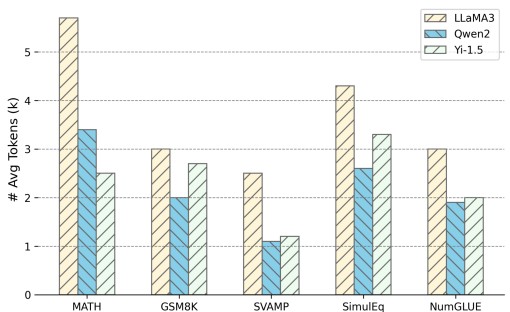

Figure 5: Average tokens needed for solving different problems.

# B    CANDIDATE ANSWER DISTRIBUTION

Figure 6 illustrates the distribution of candidate answer set sizes for individual test samples across five mathematical benchmarks (MATH, GSM8K, SVAMP, SimulEq, and NumGLUE) for three models: LLaMA3, Qwen2, and Yi-1.5. As shown in the figure, most test samples for all models tend to have larger candidate sets, with a clear peak at 12 candidates across all benchmarks. LLaMA3 consistently demonstrates larger candidate sets compared to Qwen2 and Yi-1.5, particularly in the MATH and GSM8K benchmarks, where the size of candidate sets reaches up to 12 for a substantial number of cases.

| (a) Prompt for One-Step Inference | (b) Prompt for Giving Answer | (c) Prompt for Question Disambiguation |
|---|---|---|
| Please act as a professional math teacher. Your goal is to accurately solve a math word problem. To achieve the goal, you have two jobs. # Write the NEXT step in solving the Given Question. # Do not write the full solution or final answer until prompted.

You have three principles to do this. # Ensure the solution is detailed and solves one step at a time. # Ensure each output consists of only one logical step. # Output strictly according to the format. Do not output any unnecessary content.

**Given Question:** {question}
Your output should be in the following format:
**STEP:** <your single step solution to the given question> | Please act as a professional math teacher. Your goal is to accurately solve a math word problem. To achieve the goal, you have two jobs. # Write detailed solution to a Given Question. # Write the final answer to this question. # Output strictly according to the format. Do not output any unnecessary content.

You have two principles to do this. # Ensure the solution is step-by-step. # Ensure the final answer is just a number (float or integer).

**Given Question:** {question}
Your output should be in the following format:
**SOLUTION:** <your detailed solution to the given question>
**FINAL ANSWER:** The answer is <your final answer to the question with only an integer or float number> | Please act as a professional math teacher. Your goal is to accurately clarify a math word problem by restating the question in a way that eliminates any potential ambiguity. To achieve the goal, you have two jobs. # Restate the Given Question clearly to avoid any ambiguity or confusion. # Ensure that all important details from the original question are preserved.

You have two principles to do this. # Ensure the clarified question is fully understandable and unambiguous. # Ensure that no information is lost from the original question.

**Given Question:** {question}
Your output should be in the following format:
**CLARIFIED QUESTION:** <your restated and clarified version of the original question> |

Figure 7: Prompts used in BEATS.

## C  PROMPTS

Inspired by Li et al. (2024a), we utilized the prompts shown in Figure 7 to implement the BEATS algorithm.

