# OpenReview forum: "BEATS: Optimizing LLM Mathematical Capabilities with BackVerify and Adaptive Disambiguate based Efficient Tree Search"
_ICLR.cc/2025/Conference — ICLR 2025 Conference Withdrawn Submission_

### Official Review · Reviewer_omp7 · 2024-10-30

**Soundness:** 4
**Presentation:** 3
**Contribution:** 2
**Rating:** 5
**Confidence:** 4

**Summary:**

This paper studies the mathematical reasoning problem in aspects of prompt engineering. The authors highlight the suboptimal prompts, high costs, and ineffective verification issues, and propose a tree-search-based prompt engineering method. The experiments show that the proposed method outperforms existing methods by a margin.

**Strengths:**

- The challenges proposed by the authors are reasonable. These challenges can inspire future research. The proposed method combines techniques that successfully alleviate the problems.
- The experimental results are promising. The proposed method significantly
improves the performance of each base model compared to the comparison
methods.
- This paper is well-written and organized.

**Weaknesses:**

- The novelty of this paper is somewhat limited. For example, the back verification has already been proposed in [1]. The heuristic pruning rules, e.g., Rule (3), are also common used in math reasoning. Tree-based searching methods [2] are not new either.
- The inference cost of each method should be reported. As the SFT and zero-shot methods usually require one inference, the proposed methods require multiple samplings, making the comparison unfair.
- The experimental results require deeper discussion. For example, the authors mention an issue with "ambiguous problem statements" and introduce a prompt engineering method to address it. However, there is insufficient explanation of how having the LLM rewrite the problem itself resolves this issue, and there is no comparison between the original and rewritten versions to demonstrate the effectiveness of the LLM. Additionally, if the LLM can rewrite the problem on its own, why can't it directly solve the problem?

[1] Large Language Models are Better Reasoners with Self-Verification. EMNLP
(Findings) 2023: 2550-2575

[2] Accessing GPT-4 level Mathematical Olympiad Solutions via Monte Carlo
Tree Self-refine with LLaMa-3 8B: A Technical Report.

**Questions:**

Please also refer to the weakness section.
1. The overall framework is based on prompt engineering, which strongly relies on the capability of LLM. Can the proposed method give such significant performance improvement when dealing with Olympiad math reasoning datasets, e.g., AIME, Olympiad?

---

### Official Review · Reviewer_WZRU · 2024-10-30

**Soundness:** 3
**Presentation:** 3
**Contribution:** 2
**Rating:** 6
**Confidence:** 4

**Summary:**

The paper presents BEATS, a novel approach to improving the mathematical problem-solving abilities of large language models (LLMs). It introduces a method that combines enhanced prompting, tree search with pruning, and a back-verification technique. BEATS claims significant improvements, particularly with the Qwen2-7B model, outperforming benchmarks such as GPT-4 on the MATH dataset.

**Strengths:**

- BEATS uses a unique tree search strategy with pruning and back-verification to streamline reasoning paths and verify answers, improving accuracy and efficiency.

- Empirical results across multiple datasets (MATH, GSM8K, SVAMP, etc.) show notable improvement over existing methods.

- The inclusion of a question disambiguation component helps clarify ambiguous problems, potentially reducing error.

**Weaknesses:**

- This component, though effective, adds additional steps to the inference phase, potentially affecting efficiency in real-time applications.

- The paper could benefit from a more detailed discussion of the limitations of the proposed methods and potential areas for future work, such as the impact of training data on performance.

- Further discussion on how the pruning limits affect accuracy vs. computation trade-off would add valuable insight.

**Questions:**

- How does the diversity and quality of the training data influence the performance of BEATS, particularly in edge cases or complex problems?

---

### Official Review · Reviewer_ywsT · 2024-10-31

**Soundness:** 3
**Presentation:** 4
**Contribution:** 1
**Rating:** 3
**Confidence:** 5

**Summary:**

This work investigates both prompt-based and search-based methods to enhance the mathematical reasoning abilities of large language models. The authors improve traditional search-based methods by pruning the search tree using carefully crafted prompts. A disambiguation prompt clarifies the original problem, while two additional prompts guide reasoning steps and determine search termination. Different pruning strategies are tailored to each type of prompt. The authors also introduce a self-correction mechanism called back-verification, where LLMs validate answer candidates by concatenating them with the original problem. The method’s effectiveness is evaluated across 5 math reasoning benchmarks.

**Strengths:**

1. The paper presents a novel approach that combines tree search with back-verification and adaptive disambiguation to enhance the mathematical reasoning capabilities of large language models (LLMs).
2. Ablation studies are conducted to assess the impact of key components in the proposed method, focusing on the contributions of the disambiguation and back-verification modules.
3. The pruning in the tree search effectively reduces the problem search space, improving computational efficiency.

**Weaknesses:**

1. The proposed approach lacks substantial novelty.
2. The selection of baselines for comparison in search-based methods is not sufficiently justified. Zhang et al. [1] use MCTS with LLaMA3 8B (which is also used in this paper) to enhance mathematical reasoning in LLMs, achieving 96.66% accuracy on GSM8K and 58.24% on MATH, which is significantly higher than the results of this approach.
3. Although an ablation study on the BackVerify component is included, comparisons with other verification methods are lacking. For instance, the ReST paper [2] evaluates the impact of different verifiers on performance, but similar evaluations are absent in this work.
4. While pruning tree search is a key contribution of the paper, there is no experimental analysis on the extent to which the pruning strategy reduces search time. Additionally, comparing the total inference time with other search-based methods is essential to substantiate the advantages of the pruning approach.

**References:**
- [1] Zhang, D., Huang, X., Zhou, D., Li, Y., & Ouyang, W. (2024). *Accessing GPT-4 level Mathematical Olympiad Solutions via Monte Carlo Tree Self-refine with LLaMa-3 8B*. arXiv preprint arXiv:2406.07394.
- [2] Zhang, D., Zhoubian, S., Hu, Z., Yue, Y., Dong, Y., & Tang, J. (2024). *ReST-MCTS: LLM Self-Training via Process Reward Guided Tree Search*. arXiv preprint arXiv:2406.03816.

**Questions:**

1. How do authors verify that the disambiguation prompt effectively resolves ambiguous problem statements? Although the ablation study indicates that this prompt improves final performance, a more detailed analysis is needed. For instance, do all problems correctly solved without the disambiguation prompt remain correct when it is applied?
2. Which version of GPT-4 is used for evaluation? If the results are referenced from OpenAI papers or technical blogs, please provide the appropriate citations.

---

### Official Review · Reviewer_jyaG · 2024-11-02

**Soundness:** 3
**Presentation:** 2
**Contribution:** 2
**Rating:** 3
**Confidence:** 3

**Summary:**

This paper presents BEATS, a framework that enhances mathematical problem-solving in language models by introducing targeted prompting strategies that guide the model through a step-by-step approach to decompose complex problems. Furthermore, BEATS incorporates a tree search mechanism, enabling exploration of each decision step individually, which helps refine solutions iteratively. The experiments demonstrate a significant performance increase on standard benchmarks.

**Strengths:**

The proposed method showcases two key features—*disambiguation* and *back-verification*—that notably enhance the model's reasoning process, as confirmed by the ablation study. *Disambiguation* helps clarify problem statements at each reasoning step, reducing the likelihood of misinterpretation, while *back-verification* provides a robust mechanism to cross-check each solution against previous steps. Together, these techniques improve benchmark performance by a substantial margin.

**Weaknesses:**

- The paper combines existing approaches, such as tree search and reflective reasoning techniques, but falls short of introducing transformative new methods. While effective, the design lacks substantial innovation in handling complex reasoning beyond prior approaches.

- A significant issue lies in the increased computational cost introduced by the extra steps, including disambiguation and back-verification. Although these steps improve accuracy, their contribution to computational overhead is not quantified, making it challenging to assess the overall efficiency.

- Despite mentioning computational challenges in the introduction, the paper lacks a thorough analysis of the actual cost implications. The pruning technique within tree search is minimalistic, relying on basic conditions to halt expansion rather than addressing cost at a fundamental level.

- Some areas in the paper, particularly Section 2.3, contain formatting issues, such as duplicated author names.

**Questions:**

Could the authors provide more details on the computational trade-offs involved?

---

### Note · Authors · 2024-11-13

I have read and agree with the venue's withdrawal policy on behalf of myself and my co-authors.